# Generation of a transparent killifish line through multiplex CRISPR/Cas9mediated gene inactivation

**Johannes Krug[1], Birgit Perner[1,2], Carolin Albertz[1], Hanna Mörl[1], Vera L Hopfenmüller[1], Christoph Englert[1,3]***

[1]Molecular Genetics Lab, Leibniz Institute on Aging - Fritz Lipmann Institute (FLI), Jena, Germany; [2]Core Facility Imaging, Leibniz Institute on Aging – Fritz Lipmann Institute (FLI), Jena, Germany; [3]Institute of Biochemistry and Biophysics, Friedrich-Schiller-University Jena, Jena, Germany

**Abstract** Body pigmentation is a limitation for *in vivo* imaging and thus for the performance of longitudinal studies in biomedicine. A possibility to circumvent this obstacle is the employment of pigmentation mutants, which are used in fish species like zebrafish and medaka. To address the basis of aging, the short-lived African killifish *Nothobranchius furzeri* has recently been established as a model organism. Despite its short lifespan, *N. furzeri* shows typical signs of mammalian aging including telomere shortening, accumulation of senescent cells, and loss of regenerative capacity. Here, we report the generation of a transparent *N. furzeri* line by the simultaneous inactivation of three key loci responsible for pigmentation. We demonstrate that this stable line, named *klara*, can serve as a tool for different applications including behavioral experiments and the establishment of a senescence reporter by integration of a fluorophore into the *cdkn1a (p21)* locus and *in vivo* microscopy of the resulting line.

**\*For correspondence:**
christoph.englert@leibniz-fli.de

**Competing interest:** The authors declare that no competing interests exist.

## Editor's evaluation

This important work significantly simplifies our ability to observe and manipulate aging and senescence in a living vertebrate model. The evidence supporting the conclusions is compelling, with advanced genome editing, physiological assays, and state-of-the-art microscopy. The work will be of broad interest to biomedical and aging researchers, as well as to cell biologists.

## Introduction

In animals, pigments that can be found in specific cell types limit optical transparency and prevent the *in vivo* observation of processes like organogenesis, regeneration, or cancer metastasis. While mammals have only one pigment cell type, the melanocyte, other vertebrates including fish develop several chromatophores that produce different colors. In one of the best-studied models for vertebrate coloration, the zebrafish (*Danio rerio*), the three main kinds of chromatophores are the melanophores (black), the iridophores (silvery or blue), and the xanthophores (yellow), all derived from neural crest cells (*Kelsh et al., 1996*; *Singh and Nüsslein-Volhard, 2015*). A fourth population of pigment cells, forming the retinal pigment epithelium (RPE) is derived from the optic neuroepithelium (*Bharti et al., 2006*). Different combinations of naturally occurring mutants in pigmentation genes have been used to generate adult transparent zebrafish. The *casper* line lacks melanocytes and iridophores due to mutations in *mitfa* and *mpv17*, respectively (*D'Agati et al., 2017*; *White et al., 2008*). An additional mutation in the *slc45a2* gene is present in *crystal* zebrafish, which completely lack melanin

and therefore possess a transparent RPE (*Antinucci and Hindges, 2016*). Transparent zebrafish have been used to study different aspects of cancer and stem cell biology, among others (*White et al., 2008*; *Yan et al., 2019*). In another model fish, the medaka (*Oryzias latipes*), transparent juvenile and adult animals have recently been generated through CRISPR/Cas9-mediated inactivation of *oca2* and *pnp4a* (*Lischik et al., 2019*).

During the last decade the turquoise killifish, *N. furzeri*, has emerged as a new model for research on aging (*Platzer and Englert, 2016*). With a lifespan between three and seven months, *N. furzeri* is the shortest-lived vertebrate that can be kept in captivity (*Tozzini et al., 2013*; *Valdesalici and Cellerino, 2003*). Hatchlings grow rapidly and can reach sexual maturation already within two to three weeks (*Blažek et al., 2013*). *N. furzeri* shares many hallmarks of aging with mammals, including telomere shortening, mitochondrial dysfunction, cellular senescence, loss of regenerative capacity, and cognitive decline (*Genade et al., 2005*; *Graf et al., 2013*; *Hartmann et al., 2009*; *Hartmann et al., 2011*; *Terzibasi et al., 2008*; *Valenzano et al., 2006*; *Wendler et al., 2015*). What makes the killifish an attractive model in addition, is the establishment of transgenesis and genome engineering (*Allard et al., 2013*; *Harel et al., 2015*; *Hartmann and Englert, 2012*; *Valenzano et al., 2011*) as well as the availability of reference sequences for the *N. furzeri* genome (*Reichwald et al., 2015*; *Valenzano et al., 2015*). The turquoise killifish is sexually dimorphic and dichromatic (*Cellerino et al., 2016*). Compared to females, males are larger and more colorful. The latter occurs in two color forms with red and yellow morphs that differ primarily in coloration of the caudal fin. Females have translucent fins and a pale grayish body with iridescent scales.

Here, we describe the generation of transparent juvenile and adult *N. furzeri* animals. We have used a single injection of three sgRNAs targeting *mitfa*, *ltk,* and *csf1ra*, which are involved in pigment development in melanophores, iridophores, and xanthophores, respectively. With the method employed, we have achieved simultaneous and biallelic somatic gene disruptions of three genomic loci in a highly efficient manner. Already in the F$_0$ generation, a fraction of animals were fully transparent. Homozygous triple mutants showed normal behavior, fertility, and general health. In addition, we have used the transparent line, named *klara*, to modify additional genes, to study female and male mate choice, and to generate an *in vivo* senescence reporter. For the latter, we employed homology-directed repair-mediated integration of a *GFP* allele into the locus of the senescence marker *cdkn1a* (*p21*). Finally, we show that this reporter line can be used for *in vivo* imaging.

## Results

### Multiple genes can be simultaneously inactivated in *N. furzeri*

For the generation of a transparent *N. furzeri* line, we selected the genes *mitfa*, *ltk*, and *csf1ra* as targets to interfere with the formation of melanophores, iridophores, and xanthophores, respectively. This was based on known color mutants in *D. rerio*, namely *casper* (lack of melanophores), *shady* (lack of iridophores), and *panther* (lack of xanthophores) in which *mitfa*, *ltk*, and *csf1ra* were shown to be inactivated (*Lister et al., 1999*; *Lopes et al., 2008*; *Parichy et al., 2000*). The expression of those three genes was analyzed in skin tissue from fish of both sexes at the age of 1, 2, 3, and 6 weeks posthatching (wph). In male fish, we observed a significant up-regulation of *mitfa* at the age of 6 wph, whereas *mitfa* expression did not change in female fish (*Figure 1A*). The expression of *ltk* increased steadily with age in both, male and female fish (*Figure 1B*). In contrast, *csf1ra* expression did not differ significantly over time in males and females (*Figure 1C*). Direct comparison of *mitfa*, *ltk*, and *csf1ra* expression between females and males did not reveal sex-specific differences except for *mitfa* at 6 wph (*Figure 1—figure supplement 1A–C*). To induce mutations in the selected genes, single guide RNAs were designed based on the genome sequence provided by the *N. furzeri* Genome Browser (*Reichwald et al., 2015*). Since three genes were planned to be inactivated at the same time, we used only one sgRNA per gene. To facilitate mutation detection PAM sequences were chosen that had a restriction site directly upstream, assuming that the restriction site would be lost upon the introduction of a mutation. After the characterization of different sgRNAs for each gene, sgRNAs were synthesized targeting *mitfa* in exon 6, *ltk* in exon 22, and *csf1ra* in exon 9 (*Figure 1—figure supplement 1D–F*).

To simultaneously inactivate *mitfa*, *ltk*, and *csf1ra*, we injected *Cas9* mRNA, the three different sgRNAs, and *GFP* mRNA into one-cell-stage embryos of the long-lived *N. furzeri* strain MZCS-08/122 (*Dorn et al., 2011*). *GFP* mRNA was used to identify properly injected embryos one day after the

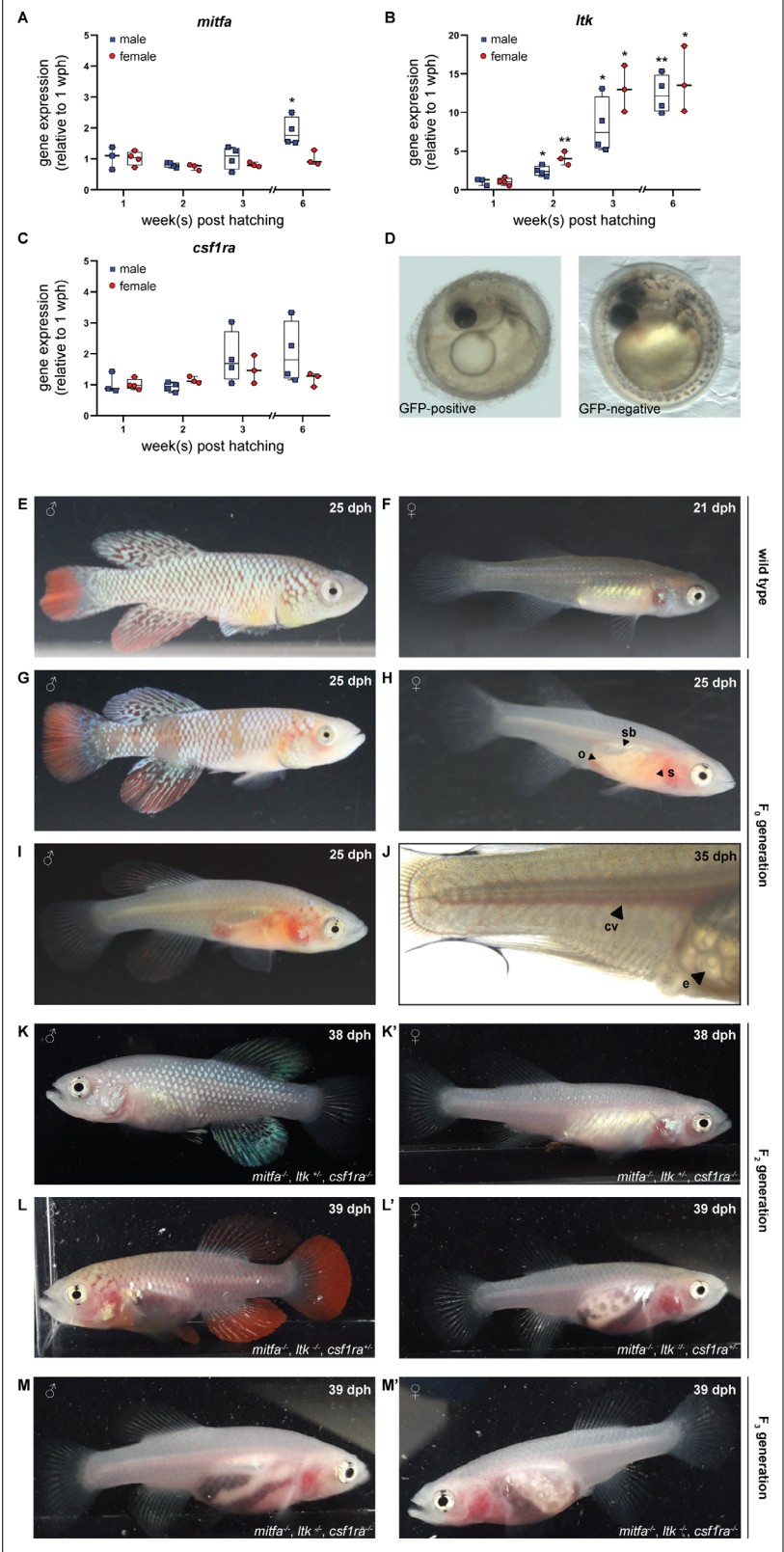

**Figure 1.** Simultaneous inactivation of genes involved in body pigmentation. (**A–C**) Gene expression analysis of *mitfa* (**A**), *ltk* (**B**), and *csf1ra* (**C**) in the skin of male and female wild-type fish via quantitative real-time PCR. The expression of *mitfa* was only significantly up-regulated in male fish at the age of 6 weeks post-hatching (wph) (p: 0.048). For both sexes, the expression of *ltk* increased significantly over time ($p_{male, 2wph}$: 0.028, $p_{male, 3wph}$: 0.026, $p_{male,}$

*Figure 1 continued on next page*

*Figure 1 continued*

6wph: 0.002; p_female, 2wph: 0.002, p_female, 3wph: 0.019, p_female, 6wph: 0.033), whereas no significant differences were observed for *csf1ra*. Expression levels were normalized to the expression at 1 wph in the respective sex. *Rpl13a* was used as housekeeping gene (n_male_1wph = 3, n_male_2wph = 4, n_male_3wph = 4, n_male_6wph = 4; n_female_1wph = 3, n_female_2wph = 4, n_female_3wph = 4, n_female_6wph = 4). Relative gene expression was calculated using the ΔΔCT method (*Pfaffl et al., 2002*). Student's or Welch's t-tests were computed to determine significant changes in gene expression. Horizontal line represents median. Whiskers show min. to max. values. (**D**) Phenotypical analysis of $F_0$ embryos revealed a reduction of melanophores in GFP-positive compared to GFP-negative embryos. (**E,F**) Male (**E**) and female (**F**) wild-type *N. furzeri*. (**G–J**) Males and females of the $F_0$ generation, display a mosaic loss of body pigmentation. Note that almost completely transparent individuals can be observed in (**H–J**), allowing a view on inner organs (o: ovary, s: stomach, sb: swim bladder). (**J**) Microscopic analysis of a female $F_0$ fish with a view on individual eggs within the ovary and blood vessels (cv: cardinal vein, e: egg). (**K,K'**) Male (**K**) and female (**K'**) fish at the age of 38 days post-hatching (dph) with the genotype *mitfa*$^{-/-}$,*ltk*$^{+/-}$,*csf1ra*$^{-/-}$ showed a lack of melanophores and xanthophores, whereas iridophores were present. (**L,L'**) A lack of melanophores and iridophores was observed in male (**L**) and female (**L'**) *N. furzeri* with the genotype *mitfa*$^{-/-}$, *ltk*$^{-/-}$, *csf1ra*$^{+/-}$. Despite a homozygous mutation in *ltk* (*ltk*$^{-/-}$) individual scales with iridophores were detected in fish of both sexes. (**M,M'**) The presence of homozygous mutations in all the three genes *mitfa*, *ltk*, and *csf1ra* resulted in the loss of body pigmentation in males (**M**) and females (**M'**) allowing a view on inner organs.

The online version of this article includes the following source data and figure supplement(s) for figure 1:

**Source data 1.** Gene expression data and N. furzeri images.

**Figure supplement 1.** Targeting the *mitfa*, *ltk*, and *csf1ra* loci in *N. furzeri*.

**Figure supplement 1—source data 1.** Gene expression data, genomic loci and microinjection.

**Figure supplement 2.** Analyzing the *mitfa*, *ltk*, and *csf1ra* loci upon simultaneous injection of three single guide RNAs (sgRNAs).

**Figure supplement 2—source data 1.** Control digest and mutation frequencies in F0 fish.

**Figure supplement 3.** Analyzing the *mitfa*, *ltk*, and *csf1ra* loci in animals of the $F_1$ and $F_2$ generation.

**Figure supplement 3—source data 1.** Sequencing results, images of F2 embryos and control digests.

---

injection. An injection mold, stabilizing the eggs during the injection procedure, has already been reported (*Harel et al., 2016*). To further improve the injection procedure, we developed a new type of injection mold having single slots for each embryo. Moreover, the wall of these slots pointing towards the direction of the injection needle is sloped facilitating the access of the needle to the egg (*Figure 1—figure supplement 1G*). Using this injection mold, we were able to inject close to 600 embryos within two days, which were sorted for a GFP signal one day post-injection. One-third of each of the embryos were GFP-positive, GFP-negative, or dead (*Figure 1—figure supplement 1H*). Seven days after injection, GFP-positive eggs were transferred onto coconut coir plates, mimicking the dry phase *N. furzeri* eggs undergo in their natural habitat. Since it is possible to detect melanophores already in embryos, GFP-positive and GFP-negative embryos were phenotypically analyzed. While melanophores were clearly detected in GFP-negative embryos, a reduction or an almost complete loss of melanophores was observed on the head and along the body axis in a fraction of GFP-positive embryos (*Figure 1D*). Iridophores and xanthophores are not detectable at this stage of development.

In order to analyze if the sgRNAs targeting *mitfa*, *ltk*, and *csf1ra* had induced mutations, regions around the expected mutation sites were amplified via PCR using DNA extracts from eight randomly selected GFP-positive and six GFP-negative embryos. Those amplicons were used in restriction enzyme digests to determine the presence of mutations. For all three genes, we observed a non-cleaved PCR fragment in all samples of GFP-positive embryos, suggesting that mutations had been introduced that made the amplicons resistant to restriction enzyme digest. The analysis of mutations in the *mitfa* sequence revealed, that 75% (6/8) of the GFP-positive samples only showed one undigested fragment, whereas 25% (2/8) were mosaic, since they also showed two additional fragments that only occur in the presence of the wild-type sequence. For *ltk*, we observed mosaicism in 37.5% of embryos (3/8) and for *csf1ra* in 87.5% (7/8) (*Figure 1—figure supplement 2A–C*). For the GFP-negative embryos, we identified digested fragments in all samples, as in the respective wild-type controls. However, the first two GFP-negative embryos also showed non-cleaved fragments, indicating the presence of mutations. This suggests that the sorting into GFP-positive and GFP-negative embryos are not fully stringent.

Next, we hatched and raised GFP-positive embryos from the $F_0$ generation. Compared to wild-type larvae and adult animals, most fish from the $F_0$ generation of successfully injected eggs showed a mosaic loss of pigment cells. However, in some of the fish we already observed an almost complete loss of pigment cells, resulting in the transparency of the animals (*Figure 1E–J*). This already allowed us to have a clear view of inner organs. At an age of 25 days post-hatching (dph), the stomach in those transparent fish showed an orange color due to nauplii of artemia, small crustaceans, which are used as food at this age. Moreover, the swim bladder and in females the ovaries were clearly visible. We also observed the blood flow in the cardinal vein and in the small vessels of the caudal fin. Additionally, we identified single follicles including lipid droplets in the ovaries. Besides this phenotypical analysis of fish from the $F_0$ generation, we also analyzed the mutation rates. Based on results from the previously mentioned restriction analysis, we observed that all 85 fish had a mutation in *mitfa*, *ltk*, and *csf1ra*, at least in a mosaic fashion. Moreover, this analysis revealed that already in the $F_0$ generation biallelic mutations were observed (*mitfa*: 48.2%, *ltk*: 67.1%, *csf1ra*: 23.5%) (*Figure 1—figure supplement 2D*). These data indicate a high efficiency of the CRISPR/Cas9 tool in *N. furzeri* and show that it is possible to simultaneously induce mutations in three genes of interest.

## Generation of a stable, transparent killifish line

For the generation of a stable, transparent *N. furzeri* line and to reduce potential off-target effects, we performed an outcross of selected $F_0$ fish with wild-type animals. As expected, all of the obtained $F_1$ progeny showed a normal pigmentation pattern and hence were phenotypically not distinguishable from wild-type animals. To assess the presence of mutations, we extracted DNA from fin biopsies for genotyping. Among 60 analyzed fish, 14 animals were triple-heterozygous ($mitfa^{+/-}$, $ltk^{+/-}$, $csf1ra^{+/-}$). The targeted loci in those fish were analyzed via sequencing. Various deletion mutations were detected, but two fish carried the same mutation at the *mitfa* (Δ11 bp), *ltk* (Δ4 bp), and *csf1ra* (Δ5 bp) locus (*Figure 1—figure supplement 3A–C*). Those animals were used for a subsequent incross, from which according to the Mendelian ratio 1/64 (1.56%) of the $F_2$ offspring was expected to be triple homozygous. We first checked whether triple homozygous embryos were viable and could be detected among the $F_2$ eggs. Hence, we randomly selected embryos to assess their genotype. We observed a lack of melanophores in a proportion of embryos. Those embryos carried a homozygous mutation in *mitfa*, whereas embryos with only a heterozygous *mitfa* mutation had melanophores (*Figure 1—figure supplement 3D*). The genotypes for *ltk* and *csf1ra* could only be assessed via molecular analysis at this developmental stage. Notably, one triple homozygous embryo was detected, indicating that at least until the hatching stage those embryos were viable (*Figure 1—figure supplement 3E and F*). We then proceeded with the hatching of $F_2$ eggs. In this generation, different genotypes were observed to result in different phenotypical appearances. Female fish with a heterozygous mutation in *ltk* and homozygous mutations in *mitfa* and *csf1ra* resembled wild-type females at first glance, whereas in male fish especially the lack of xanthophores resulted in a silver-blue appearance (*Figure 1K and K'*). In contrast to this, the lack of melanophores and iridophores in fish with the genotype $mitfa^{-/-}$, $ltk^{-/-}$, $csf1ra^{+/-}$ allowed a view on inner organs, particularly in females (*Figure 1L and L'*). In order to increase the likelihood to obtain transparent, triple homozygous fish, we crossed two fish with the genotype $mitfa^{-/-}$, $ltk^{-/-}$, $csf1ra^{+/-}$. From this cross, we genotyped 50 individuals via high-resolution melting analysis (HRMA) and identified 13 triple homozygous fish. We named the resulting transparent *N. furzeri* line *klara* (*Figure 1M and M'*).

## Characterization of *klara* animals

To assess the health and survival of the *klara* line, we compiled data from our fish database, in which the date of birth and date of death for every fish is recorded. The data of the wild-type and *klara* fish that were hatched in 2020 and 2021 comprise 546 and 159 animals, respectively. The corresponding survival curves did not show any differences between the cohorts (*Figure 2—figure supplement 1A*). This is supported by the fact that both *klara* and wild-type animals score indifferently upon daily inspection of swimming, feeding, and social behavior as well as body condition, appearance, and breathing. Finally, we have quantified size and weight until 5 weeks post-hatching in both cohorts. Except for a single time point in females, there were no significant differences between the cohorts (*Figure 2—figure supplement 1B*). These data suggest that in terms of general physiology, *klara* animals are not different from wild-type fish.

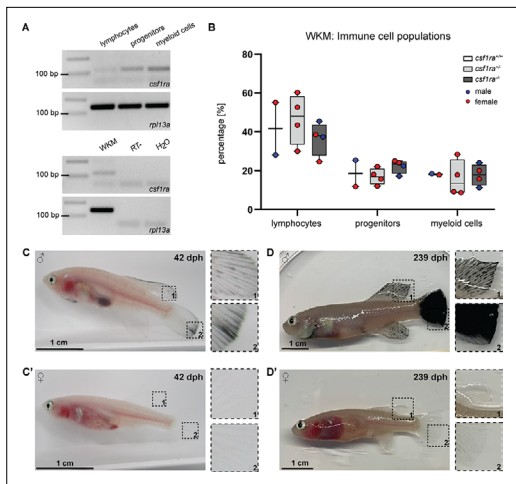

**Figure 2.** Characterization of *klara*. (**A**) Expression of *csf1ra* was analyzed via RT-PCR using cDNA from FACS-sorted populations of lymphocytes, progenitors, and myeloid cells obtained from the whole kidney marrow (WKM) of a wild-type *N. furzeri*. Csf1ra was detected in all subpopulations, most strongly in myeloid cells. As a negative control, an RT-sample (no reverse transcriptase during cDNA synthesis) was used to exclude contaminations with genomic DNA. As loading control, *rpl13a* was used. (**B**) Comparison of cell numbers in the different subpopulations of the WKM of fish with the following genotypes: *mitfa⁻/⁻, ltk⁻/⁻, csf1ra⁺/⁺* (n=2), *mitfa⁻/⁻, ltk⁻/⁻, csf1ra⁺/⁻* (n=4), and *mitfa⁻/⁻, ltk⁻/⁻, csf1ra⁻/⁻* (n=4). One-way ANOVA followed by Tukey's post hoc test did not reveal any significant differences. Horizontal line represents median. Whiskers show min. to max. value. (**C,C'**) Male *klara* fish (**C**) showed an appearance of melanophores on fin appendages, which was not observed in females (**C'**). (**D,D'**) Occurrence of melanophores intensified with age resulting in black fins in male fish (**D**). In female *klara* animals (**D'**) black fins were not observed.

The online version of this article includes the following source data and figure supplement(s) for figure 2:

**Source data 1.** PCR, FACS analysis of the WKM and images of klara fish.

**Figure supplement 1.** Characterization of *klara* fish.

**Figure supplement 1—source data 1.** Lifespan data, size and weight measurements and gating strategy.

In zebrafish, the role of *csf1ra* in xanthophore development has been described (*Parichy et al., 2000*). In addition, *csf1ra* is also known to play a role in the immune system, in particular in the survival, proliferation, and differentiation of monocytes and macrophage (*Hume et al., 2017*; *Tagoh et al., 2002*). For this reason, we wondered whether the inactivation of *csf1ra* has an effect on the immune cell population of *klara*. Since the kidney is the primary hematopoietic organ in teleost fish, we analyzed the whole kidney marrow (WKM) of fish with homozygous mutations in *mitfa* and *ltk*, which had in addition either no mutation, a heterozygous or a homozygous mutation in *csf1ra*. Based on a published gating strategy (*Sanz-Morejón et al., 2019*), we identified four subpopulations in the WKM of *N. furzeri* using flow cytometry (*Figure 2—figure supplement 1C and D*). The strongest *csf1ra* expression was detected in the myeloid cell population, which contains macrophages (*Figure 2A*). Comparing the number of immune cells in the three subpopulations, we did not observe any differences among the *csf1ra* genotypes (*Figure 2B*).

During the raising of *klara* fish, we observed that around the age of approximately four weeks, i.e., at the time of sexual maturation, melanophores appeared in male fish, in particular on fin appendages. However, this was not detected in female *klara* fish (*Figure 2C and C'*). With age, the melanophores spread over the whole fins and resulted e.g., in a fully black caudal fin (*Figure 2D and D'*). This, however, did not interfere with the overall body transparency of *klara* animals.

## Males and females prefer pigmented mating partners

Besides its role as camouflage, protection from UV damage, or for recognition, body pigmentation also plays an important role in the choice of mating partners. We investigated whether the lack of body pigmentation affects the breeding behavior of *klara* fish. We set up three breeding groups consisting of one *klara* male and two *klara* females, at the age of 14 weeks, which had never been used for breeding before. *Klara* fish showed a normal mating behavior, whereby the male uses its caudal fin to push the female into the sand and thus induces egg laying (*Figure 3—video 1* and *Figure 3—video 2*). We also assessed the quantity and quality of collected eggs, which did not differ from wild-type fish, so that we could maintain the *klara* line in a triple homozygous state (*Figure 3A*). Thus, *klara* and wild-type females do not show a difference in fecundity. To assess the role of body pigmentation for mate choice in killifish, we set up different combinations of breeding trios consisting of wild-type and *klara* fish, thus that a wild-type or a *klara* animal of each sex had the choice between a wild-type or a *klara* animal of the other sex (*Figure 3B*). For each of the four combinations, we analyzed three tanks. Fish were put together and a sand box, which is required for egg deposition, was put into the tank. After 10 days

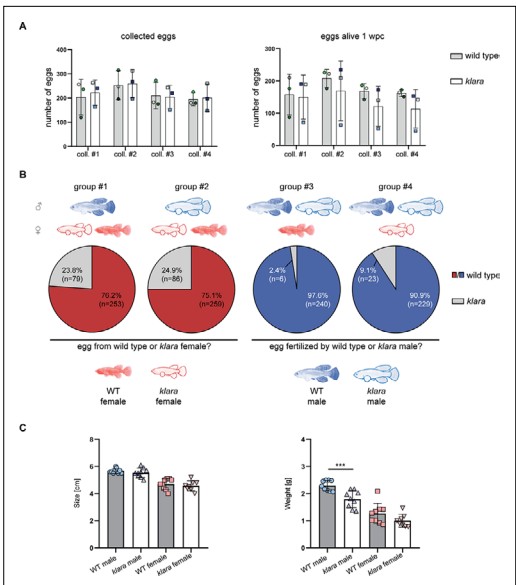

**Figure 3.** Pigmented fish are preferred mating partners. (**A**) For the analysis of egg quantity, eggs were collected once per week for four weeks (coll. #1–4) from wild-type (n=3 tanks with one male and two females each) and *klara* (n=3 tanks with one male and two females each) fish. To assess egg quality, the number of alive eggs stored on coconut coir plates was determined one-week post-collection (wpc). Student's or Welch's t-tests were computed revealing no significant differences regarding egg quantity and quality. Error bars represent standard deviation. (**B**) Collected eggs obtained from breeding groups with a wild-type female and a *klara* female and either a wild-type male (group #1) or a *klara* male (group #2) were genotyped via HRMA. As a mating partner, male fish preferred wild-type females (group #1: 76.2%; group #2: 75.1%) over *klara* females (group #1: 23.8%; group #2: 24.9%). In the presence of a wild-type male and a *klara* male and either a wild-type female (group #3) or a *klara* female (group #4) the majority of analyzed eggs were fertilized by the wild-type male (group #3: 97.6%; group #4: 90.9%). (**C**) Within the same sex, wild-type and *klara* fish did not differ in size($n_{WT\_male}$ = 9 $n_{WT\_female}$ = 9, $n_{klara\_male}$ = 9, $n_{klara\_female}$ = 8). Wild-type males were significantly heavier than *klara* males (p: 0.0009). Student's or Welch's t-tests were computed to determine differences in size or weight. Error bars represent standard deviation.

The online version of this article includes the following video and source data for figure 3:

**Source data 1.** Egg quantity and quality data, genotyping of eggs and size and weight measurements.

**Figure 3—video 1.** Mating of wild-type *N. furzeri* animals.

https://elifesciences.org/articles/81549/ figures#fig3video1

**Figure 3—video 2.** Mating of *N. furzeri klara* animals.

*Figure 3 continued on next page*

https://elifesciences.org/articles/81549/ figures#fig3video2

the sand box was removed for the following two days. Subsequently, the box was added again and for the following four weeks, we collected eggs once per week for further analysis. To identify who produced the egg or who had fertilized it, we determined the genotype of the fertilized eggs by HRMA. We observed that in the presence of a *klara* and a wild-type female fish, irrespective of whether the male was a wild-type or a *klara* animal, approximately 75% of fertilized eggs originated from the wild-type female (*Figure 3B*). This indicated that both *klara* and wild-type males showed a preference for the pigmented wild-type female. This mate choice was not influenced by size or weight, since both parameters were not statistically different between wild-type and *klara* females (*Figure 3C*). Similarly, in the breeding groups, in which a *klara* and a wild-type male were present, more than 90% of eggs were fertilized by the wild-type male (*Figure 3B*). Again, wild-type and *klara* males did not show a difference in size, although *klara* males had less weight (*Figure 3C*). Taken together, this competitive breeding experiments indicated that pigmented fish were the preferred mating partner for both sexes.

## *Klara* fish can serve as a platform for further genetic manipulation

The simultaneous inactivation of *mitfa*, *ltk*, and *csf1ra* resulted in a loss of body pigmentation, however, the eyes were still normally pigmented (*Figure 4A*). Transparent zebrafish from the *casper* line (*White et al., 2008*) also still have pigmented eyes, while fish of the fully transparent *crystal* line lack those pigments. This was achieved via an inactivation of the *slc45a2* gene (*Antinucci and Hindges, 2016*). To get rid of retinal pigmentation in *klara* animals we designed a sgRNA targeting the *slc45a2* locus (*Figure 4—figure supplement 1A*). We first tested this sgRNA in one-cell stage eggs from the wild-type strain. Compared to GFP-negative embryos, we observed a loss of pigmentation in the eye of GFP-positive embryos (*Figure 4B*). Notably, also the appearance of melanophores on the head and body of those embryos was reduced. This phenotypical analysis indicated the inactivation of *slc45a2*, which was subsequently confirmed via restriction enzyme digest (*Figure 4—figure supplement 1B*). We then performed microinjections using the sgRNA targeting *slc45a2* into *klara* one-cell stage

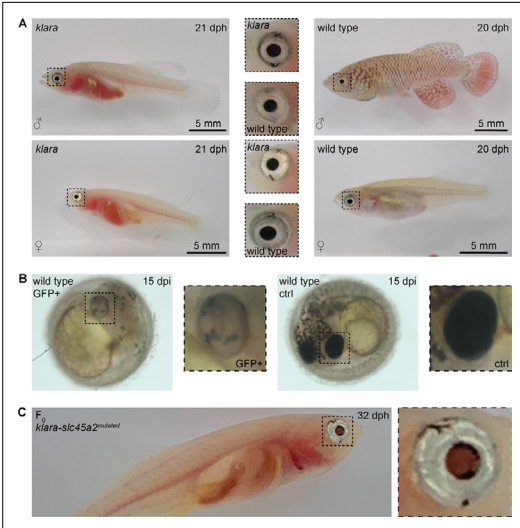

**Figure 4.** Partial loss of eye pigmentation. (**A**) Despite the inactivation of three genes involved in pigmentation, the eye pigmentation did not differ between *klara* and wild-type fish. (**B**) Targeting the *slc45a2* locus in wild-types resulted in a reduction of melanophores in the eye, but also on the whole body of injected embryos compared to wild-type controls. (**C**) A loss of melanophores in the retinal pigmented epithelium was observed in $F_0$ fish upon a microinjection of the sgRNA targeting *slc45a2* in *klara* embryos. A silver-pigmented ring around the eye was still present. (ctrl: control; dpi: days post-injection; dph: days post-hatching).

The online version of this article includes the following source data and figure supplement(s) for figure 4:

**Source data 1.** Images of embryos and fish.

**Figure supplement 1.** Inactivation of the *slc45a2* locus.

**Figure supplement 1—source data 1.** slc45a2 locus, control digests and sequencing results.

embryos and hatched $F_0$ fish (n=7). Besides the transparent body, the black pigmentation of the eye was absent, while we still observed a silver-pigmented ring around the eye of the animals (*Figure 4C*). The presence of a mutation in the *slc45a2* locus of all seven $F_0$ fish was confirmed via restriction enzyme digest (*Figure 4—figure supplement 1C*). Subsequently, we performed an outcross of $F_0$ fish with *klara* and identified the presence of various indel mutations in $F_1$ offspring (*Figure 4—figure supplement 1D and E*). This experiment showed that the *klara* line can serve as a platform for further genetic manipulations.

## Generation of an *in vivo* senescence reporter

The accumulation of senescent cells is one of the hallmarks of the aging process (*López-Otín et al., 2013*). We wanted to take advantage of the transparent *klara* line and generate a senescence reporter line. To this end, we inserted a reporter construct, consisting of a *GFP* and *nitroreductase* (*NTR*) cassette into the *cdkn1a* (*p21*) locus of *klara* via homology-directed repair (*Figure 5A*). The *cdkn1a* gene is a senescence marker and upregulated in old killifish (*Graf et al., 2013*). With this reporter/NTR cassette, *cdkn1a* expressing cells can be labeled via GFP and can also be ablated via the NTR/Mtz system (*Curado et al., 2007*). To facilitate HDR, we added flanking arms of 903 bp and 901 bp on the 5' and 3' ends of the construct, respectively. Since it has been reported that 5'-modifications of double-stranded donor templates increase HDR efficiency (*Ghanta et al., 2021*; *Gutierrez-Triana et al., 2018*), we amplified the template using biotinylated oligonucleotides (*Figure 5B*). Subsequently, we performed micro-injections into one-cell stage *klara* eggs using an injection solution with a sgRNA that induces a DNA double-strand break in close proximity to the intended insertion site and the biotinylated HDR donor template. In 1 out of 16 randomly selected GFP-positive (i.e. successfully injected) embryos, we detected the presence of the reporter cassette. Its proper insertion was verified via sequencing (*Figure 5—figure supplement 1A*). We subsequently set up the remaining GFP-positive embryos for hatching and detected three fish with a proper insertion among 19 $F_0$ animals. To verify the proper insertion of the *GFP/ntr* cassette into the genome of *klara* animals and for subsequent genotyping, we employed PCR using two different pairs of primers (*Figure 5—figure supplement 1B*).

Next, we performed an outcross of the $F_0$ animals with *klara* fish. To assess whether the reporter was functional, the obtained $F_1$ embryos were exposed to a γ–irradiation dosage of 10 Gy. This should induce DNA damage and lead to the activation of TP53 and up-regulation of *cdkn1a*. To analyze, whether the up-regulated *cdkn1a* expression is linked to an up-regulation of *GFP*, we analyzed the embryos 1 hr before as well as 24 hr post irradiation (hpi) via fluorescence microscopy (*Figure 5—figure supplement 2A*). Before irradiation, autofluorescence originating from the yolk was observed in *cdkn1a*$^{+/+}$ and *cdkn1a*$^{ki/+}$ embryos (*Figure 5C*). In contrast, at 24 hpi we observed GFP-positive cells in the optic tectum and other parts of the developing brain and fin buds of *cdkn1a*$^{ki/+}$ embryos (*Figure 5D*). Gene expression analysis revealed an upregulation of endogenous *cdkn1a* as well as both

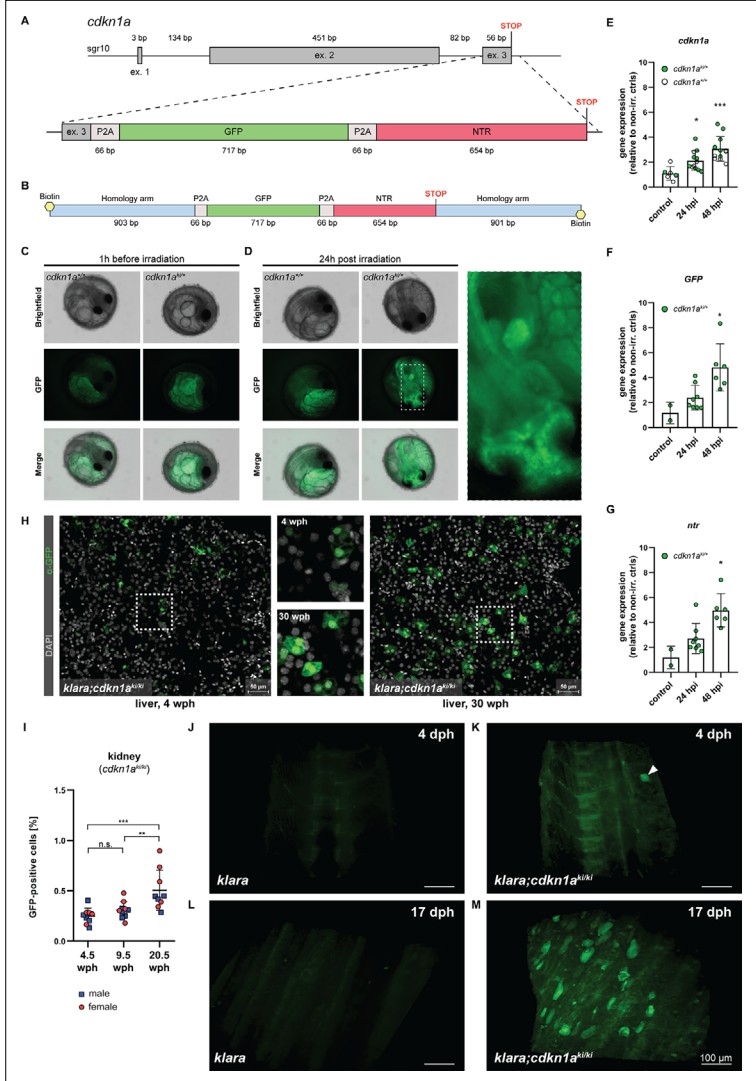

**Figure 5.** CRISPR/Cas9-mediated insertion of a reporter construct. (**A**) The *cdkn1a* locus in *klara* was targeted via CRISPR/Cas9 to insert a reporter construct consisting of a GFP and an NTR cassette, separated by P2A self-cleavage sites, allowing the detection and ablation of senescent cells. (**B**) A 5′-biotinylated, double-stranded donor template flanked by two ~0.9 kb flanking arms was used for the target-specific insertion. (**C,D**) Function of the reporter construct was tested by exposing embryos (*cdkn1a*$^{+/+}$ and *cdkn1a*$^{ki/+}$) to a γ-irradiation dose of 10 Gy. Representative images taken before irradiation showed autofluorescence originating from the yolk, which was observed in embryos of both genotypes (**C**). 24 hr post-irradiation (hpi), the presence of GFP-positive cells was detected only in *cdkn1a*$^{ki/+}$ embryos (**D**). (**E**) Gene expression analysis of *cdkn1a* in embryos (*klara;cdkn1a*$^{+/+}$ and *klara;cdkn1a*$^{ki/+}$) at 24 (n=12) and 48 hpi (n=12) compared to non-irradiated controls (n=6). Expression of *cdkn1a* increased significantly as revealed by Student's t-test ($p_{24hpi}$: 0.011, $p_{48hpi}$: 0.0004). (**F,G**) For the expression of *GFP* (**F**) and *ntr* (**G**) at 24 (n=8) and 48 hpi (n=6) compared to non-irradiated controls (n=2), *klara;cdkn1a*$^{+/+}$ embryos were excluded due to the absence of the reporter construct. Expressions of *GFP* and *ntr* were significantly increased at 48 hpi ($p_{GFP}$: 0.045, $p_{ntr}$: 0.006) compared to controls. For all qRT-PCRs, *rpl13a* was used as a housekeeping gene. Relative gene expression was calculated using the ΔΔCT method. Student's t-tests were computed to determine significant changes in gene expression. Error bars represent standard deviation. (**H**) Detection of GFP-positive cells in the liver of a young (4 wph) and an old (30 wph) *klara;cdkn1a*$^{ki/ki}$ fish. A higher number of GFP$^+$ cells was detected in the old liver sample. Images are displayed as extended depth of focus projections. In the young liver tissue, small individual GFP-positive cells were detected, whereas in the old liver, a high prevalence of enlarged GFP-positive cells was identified (see zoom-in). (**I**) Analysis of kidney samples via flow cytometry from *klara;cdkn1a*$^{ki/ki}$ fish at the age of 4.5 wph (n$_{male}$ = 5; n$_{female}$ = 4), 9.5 wph (n$_{male}$ = 5; n$_{female}$ = 5) and 20.5 wph (n$_{male}$ = 4; n$_{female}$ = 5). A significantly higher proportion of GFP-positive cells was detected in 20.5 wph versus 4.5 wph (p: 0.0009) and 9.5 wph (p: 0.0084) fish. One-Way ANOVA followed by Tukey's post hoc test was computed to analyze for differences

*Figure 5 continued on next page*

*Figure 5 continued*

in cell number. Horizontal line shows mean value. Error bars represent standard deviation. (**J–M**) Representative images from light sheet microscopy show the presence of GFP-positive cells in living *klara;cdkn1a*$^{ki/ki}$ fish (**K, M**) at 4 dph (**K**) and 17 dph (**M**). The panel displays representative 3D reconstructions from stacks of 31 (**J,L**) and 372 (**K,M**) optical slices. *Klara* fish at the same ages (**J,L**), which do not have any GFP + cells, were used as controls. The image processing settings for the *klara* fish were adjusted to make the background fluorescence visible in order to create comparable image pairs. While a high number of GFP-positive cells was present in the dorsal fin area of *klara;cdkn1a*$^{ki/ki}$ fish at the age of 17 dph (**M**), only one cell (marked by the white arrowhead) was detected in the dorsal developing fin of the younger *klara;cdkn1a*$^{ki/ki}$ animal (**K**).

The online version of this article includes the following video, source data, and figure supplement(s) for figure 5:

**Source data 1.** cdkn1a locus, images of embryos, gene expression data, immunofluorescence stainings, FACS analysis and images obtained by light sheet microscopy.

**Figure supplement 1.** Analysis of embryos from the senescence reporter line.

**Figure supplement 1—source data 1.** Genotyping PCRs and sequencing results.

**Figure supplement 2.** Characterization of the senescence reporter line.

**Figure supplement 2—source data 1.** Schematic presentation of workflows and gene expression data.

**Figure 5—video 1.** This movie represents the z-stacks of optical slices from which the 3D projections of *Figure 5K* were generated.

https://elifesciences.org/articles/81549/figures#fig5video1

**Figure 5—video 2.** This movie represents the z-stacks of optical slices from which the 3D projections of *Figure 5M* were generated.

https://elifesciences.org/articles/81549/figures#fig5video2

*GFP* and *ntr* expression upon γ–irradiation in *cdkn1a*$^{ki/+}$ embryos, which further significantly increased at 48 hpi (*Figure 5E–G*). Thus, we were able to insert a functional 1.5 kb reporter cassette into the *cdkn1a* locus of *klara* fish.

To verify the functionality of the *nitroreductase* gene, we employed cell culture (*Figure 5—figure supplement 2B*). We isolated skin cells from adult fish of the *cdkn1a*-reporter *klara* line. Several weeks after the primary cells had been taken in culture, they were treated with 50 µM metronidazole (Mtz) and subsequently exposed to 10 Gy γ–irradiation. After another 24 hr cells were harvested, and RNA was isolated and analyzed by qRT-PCR. We observed a consistent upregulation of *cdkn1a*, *GFP* as well as *ntr* upon irradiation (*Figure 5—figure supplement 2C*). This upregulation was reversed by Mtz treatment. Although the data did not reach statistical significance, these results suggest that the *ntr* cassette is functional and leads to the decay of cells upon Mtz application.

After the initial characterization of the *cdkn1a*$^{ki/+}$ embryos, we raised them to adulthood and generated F$_2$ offspring. Subsequently, we used immunofluorescence and flow cytometry to detect and quantify the occurrence of GFP$^+$ cells and their potential accumulation upon aging. As organs for analysis, we selected liver and kidneys from young and old *cdkn1a*$^{ki/ki}$ animals. Staining for GFP revealed many more GFP$^+$ cells in the liver of a 30-weeks-old *cdkn1a*$^{ki/ki}$ fish than in a 4-weeks-old animal (*Figure 5H*). Of note, GFP$^+$ cells in older tissue were larger and often appeared clustered as opposed to smaller and scattered GFP$^+$ cells in young tissue. Using FACS, we detected a significant increase of GFP$^+$ cells in 20.5-weeks-old kidneys, compared to 4.5-and 9.5-weeks-old animals (*Figure 5I*).

Finally, we performed light sheet microscopy on the dorsal fin area of live *cdkn1a*$^{ki/ki}$ *klara* fish at 4 dph (*Figure 5K* and *Figure 5—video 1*) and 17 dph (*Figure 5M* and *Figure 5—video 2*) with *klara* animals as controls (*Figure 5J and L*). While a significant number of GFP-positive cells was detected at the older stage, only a single cell was observed in the 4-days-old animal. These data suggest that the reporter that we have generated can be used to monitor *cdkn1a (p21)*-positive cells *in vivo* and that these cells accumulate upon aging.

## Discussion

Here, we report the generation of a transparent killifish line that is lacking melanophores, iridophores, and xanthophores. Based on published literature, we have selected the three genes *mitfa*, *ltk,* and *csf1ra* as targets for CRISPR/Cas9-mediated inactivation in *N. furzeri* (*Lister et al., 1999*; *Lopes et al.,*

*2008*; *Parichy et al., 2000*). We employed a single injection with three sgRNAs that had been pre-selected and characterized. The observation that some of the injected $F_0$ embryos showed a complete loss of melanophores demonstrated that the CRISPR/Cas9 system acts very efficiently in *N. furzeri*. This has been observed before (*Harel et al., 2015*; *Oginuma et al., 2022*) and is most likely explained by the long duration of the one-cell stage in this species of 2–3 hr during which Cas9 can act (*Dolfi et al., 2014*). Cas9 efficiency was further confirmed by the fact that for all three genes, both alleles were inactivated in the $F_0$ animals, many of which showed an almost complete transparency. This efficiency is comparable to the one reported for zebrafish, whereby a codon-optimized Cas9 protein was employed to target a reporter transgene and four endogenous loci. In this case, mutagenesis rates reached 75–99% (*Jao et al., 2013*).

The *klara* animals did not show any anomalies regarding general health, lifespan, phenotype, and behavior and could be bred to homozygosity and kept as a stable line. While *klara* animals were initially fully transparent, male animals developed black pigments, particularly on fin appendages, which increased during their lifespan. While the re-appearance of pigmentation in male fin appendages limits the application of this line in the adult stages, it suggests that there is a second, *mitfa*-independent population of melanophores in killifish that appears in later life. In zebrafish, it has been shown that the paralogous gene, *mitfb* might fulfill this role by activating *tyrosinase* expression (*Lister et al., 2001*). Whether or not this is also the case in killifish and whether *mitfb* might also be responsible for the remaining pigmentation in the retina remains to be determined. By inactivating *slc45a2* in *klara* fish, we obtained animals that were fully transparent regarding their body and the eye. This quadruple mutant might be particularly interesting for research on eye and retina regeneration and shows that *klara* animals can be used as a background for further gene inactivation.

We used the *klara* line for addressing mate choice in killifish. While breeding behavior was normal among *klara* animals, both wild-type and *klara* animals preferred pigmented mating partners in competitive breeding situations. Here, mate choice for pigmented partners was more pronounced in females (>90%) than in males (approx. 75%). As we showed that there is no difference in fecundity between *klara* and wild-type females, this is most likely explained by the difference in appearance between wild-type and *klara* animals. This difference is much more distinct between males than between the respective females. The choice for pigmented males might also be influenced by the larger weight of wild-type males. As male mating success is driven by dominance, which is influenced by weight, it is likely that wild-type males have an advantage over *klara* males in getting access to females and their eggs. In this case, the strong selection for wild-type males would result from both female selection and male-dominant behavior.

Our observation is in line with an earlier report on the two-spotted gobies. In this case, males preferred to mate with more colorful females, which have bright yellow-orange bellies during the breeding season (*Amundsen and Forsgren, 2001*). It is very surprising that for mate choice *N. furzeri* seems to rely on visible cues, as their natural habitat is turbid with limited visibility (*Reichard and Polačik, 2019*). It is tempting to speculate that mate choice in *N. furzeri* could also be influenced by other traits like chemical signals including pheromones.

The main motivation to generate a transparent killifish line was the possibility to perform longitudinal studies regarding aging and regeneration and to be able to observe respective processes in real-time without having to sacrifice cohorts of animals at distinct time points. One of the hallmarks of aging is the accumulation of senescent cells that are characterized by the expression of specific markers including *cdkn1a* (*p21*) and *cdkn2a* (*p16*) (*López-Otín et al., 2013*). It is still a matter of debate whether senescent cells limit or extend lifespan (*Grosse et al., 2020*; *van Deursen, 2014*). To visualize and address the role of senescent cells, we have integrated a cassette encompassing a fluorescence reporter as well as a nitroreductase allele into the *cdkn1a* locus of *klara* animals. The respective HDR template was flanked by 0.9 kb homology arms and carried biotinylated 5'-ends, as those modifications of double-stranded DNA donor templates have been reported to increase HDR efficiency (*Ghanta et al., 2021*; *Gutierrez-Triana et al., 2018*). Out of 35 injected embryos, four showed proper integration of the construct, corresponding to 11%. At least three $F_0$ animals passed on the engineered allele to the next generation. Our analysis of *klara* embryos harboring the *GFP* allele in the *cdkn1a* locus had shown that the integrated reporter is functional and can be activated upon γ-irradiation. The characterization of respective adult animals demonstrated an accumulation of $GFP^+$ cells with age, as determined by flow cytometry, immunofluorescence, and *in vivo*

imaging. Whether those cells are truly senescent cells remains to be determined. Since the reporter also encompasses an *ntr* allele, it shall be possible to ablate the GFP⁺ and thus presumably senescent cells by administration of metronidazole, a prodrug that is converted to a cytotoxic agent by NTR. With the transparent senescence reporter line, it will be possible to further characterize the function of senescent cells in development, aging, and regeneration. The generation of additional reporters into different loci, e.g., *cdkn2a/b* or other senescent markers will also allow addressing the possible heterogeneity of senescent cells (*Cohn et al., 2023*).

We consider the *klara* fish that we describe here a valuable and versatile tool for research on aging, regeneration, and behavior. This fish line will also be beneficial for colleagues interested in cancer biology and ecology. Beyond its potential to be used for the investigation of questions in biology *klara* animals can also contribute to the reduction of animal numbers, an aspect that gains increasing importance in biomedical research.

## Materials and methods
### Fish husbandry
All the work reported here was performed in the background of the wild-type *N. furzeri* strain MZCS-08/122, which is originally derived from southern Mozambique (*Dorn et al., 2011*), or the *klara* line. Fish are kept in single-housing at 26 °C on a light:dark cycle of 12 hr each. Adult fish are fed once a day *ad libitum* with red mosquito larvae, whereas juvenile fish (up to 5 weeks post-hatching) are fed with nauplii of artemia twice a day. To obtain a high number of fertilized wild-type oocytes for injections, multiple breeding groups consisting of ten fish ( two males, eight females) were set up in 40 liter tanks. To obtain oocytes from *klara* fish, breeding groups of one male and three females were set up in tanks containing approximately 8.5 l tank water. The sand box, which is necessary for the deposition of eggs, was always removed two days before and put back into the tank 2 hr before the injection. The eggs were collected with a sieve and were then used for microinjections. The routine collection of eggs for line maintenance was done on a weekly basis. Eggs were put on coconut coir plates and stored at 29 °C.

All fish were maintained in the Nothobranchius facility of the Leibniz Institute on Aging – Fritz Lipmann Institute Jena according to the German Animal Welfare Law. All performed experiments were covered by the animal licenses FLI-17–016, FLI-20–001, and FLI-20–102, which were approved by the local authorities (Thüringer Landesamt für Verbraucherschutz).

### Lifespan analysis
The data for the lifespan of wild-type and *klara* fish were not obtained from a dedicated lifespan experiment. Those data were extracted from our fish database (a-tune tick@lab), in which date of birth and date of death are recorded. In the shown Kaplan-Meier curve, all wild-type and *klara* fish that hatched in 2020 and 2021 in our facility were included. Fish, which had to be euthanized due to human endpoint criteria or died already during raising before reaching 28 dph, were excluded.

### Growth and weight measurements
The assessment of body size and body weight were performed on freshly euthanized fish (wild-type and *klara*) at the age of 7, 14, 21, 28, and 35 days post-hatching. Fish were briefly dried with a tissue and then placed onto a scale to determine body weight. Subsequently, fish were transferred onto a petri dish and placed on a millimeter of paper to assess body length. Body length was measured from the tip of the mouth to the end of the caudal fin. Since the sex of young fish cannot be distinguished phenotypically, a fin biopsy was taken from each fish as soon as the measurements were completed. This tissue sample was used for a PCR to identify the sex. The PCR was performed as described (*Richter et al., 2022*).

### Design and synthesis of single guide RNAs (sgRNAs)
SgRNAs were designed based on the genome sequence provided by the *N. furzeri* Genome Browser (*Reichwald et al., 2015*). Target sequences for sgRNAs had a length of 20 nucleotides followed by the PAM sequence -NGG. Only sequences containing a restriction site directly upstream of the PAM sequences were selected. A TAGG-overhang was added to the 5'-end of the forward sgRNA

oligonucleotide and an AAAC-overhang to the 5'-end of the reverse complementary oligonucle-otide (sg_mitfa_1: 5'-TAGG-AACATCAAAAGGGAATTCAC-3' sg_mitfa_2: 5'-AAAC- GTGAATTC CCTTTTGATGTT-3', sg_ltk_1: 5'- TAGG-TGAAATGGATTTCCTGATGG-3' sg_ltk_2: 5'-AAAC-CCAT CAGGAAATCCATTTCA-3', sg_csf1ra_1: 5'-TAGG-CAGAGACACTTTTTCCATGG-3' sg_csf1ra_2: 5'-AAAC-CCATGGAAAAAGTGTCTCTG-3', sg_slc45a2_1: 5'- TAGG-TGACTACTGCCGCTCACAGT -3' sg_slc45a2_2: 5'- AAAC-ACTGTGAGCGGCAGTAGTCA-3'). Complementary sgRNA oligonucle-otides were annealed by heating them up to 95°C followed by gradually cooling by 1°C per 30s. The annealed oligonucleotides were ligated into the *BsaI*-linearized pDR274 vector (Addgene, plasmid #42250). After the ligation, *E. coli* TOP10 cells were transformed with this vector, containing the sgRNA sequence. Isolated plasmids were checked via sequencing for the correct presence of the sgRNA sequence. Using the *DraI* restriction enzyme a fragment of approximately 300bp was excised from the plasmid containing the sgRNA and the T7 promoter sequence. This fragment was used as a template for the *in vitro* transcription, which was performed according to the manufacturer's protocol of the mMESSAGE mMACHINE T7 Transcription Kit (Thermo Scientific Inc). The quality of *in vitro* transcribed sgRNAs was controlled via RNA agarose gel electrophoresis.

## Design and synthesis of DNA donor templates for HDR

The assembly of the donor template for the insertion of a *P2A-GFP-P2A-NTR* cassette into the *cdkn1a* locus of *klara* was done using the *NEBuilder HiFi DNA Assembly Cloning Kit*. P2A sites were added to the *GFP* (derived from the *tol2* kit plasmid #395) and the *NTR* sequence (obtained from plasmid: *Myl7-LoxP-myctagBFP-LoxPNTRmCherry Zhou and Hildebrandt, 2012*) via PCR. The *NEBuilder Assembly Tool* was used for the design of oligonucleotides containing overlap sequences, which are required for the assembly of individual PCR fragments. 0.05 pmol of each fragment (flanking arms, *P2A-GFP*, *P2A-NTR*, pGGC (pUC57-BsaI) backbone vector *Geissler et al., 2011*) were used together with 10μl of the *NEBuilder HiFi DNA Assembly Master Mix*. The NEBuilder assembly reac-tion was performed for 1 hr at 50°C. 5 μl of this reaction was used for a subsequent transformation into chemically competent TOP10 *E. coli*. Isolated plasmids were checked via sequencing for correct template assembly. 5'-biotinylated oligonucleotides (bio_cdkn1a_fw: 5'-TCTTACACCAAACACC ACAA-3' bio_cdkn1a_rv: 5'-TAAAACATGCAGGATACCGG-3') were used to amplify the template from the isolated and then linearized plasmid. The amplicon with the expected size was excised from the agarose gel and purified using the *NucleoSpin Gel and PCR clean-up* kit (Macherey-Nagel). To induce a DNA double-strand break in close proximity to the intended site of insertion, the following oligonucleotides for sgRNA synthesis were used: sg_cdkn1a_1: 5'-TAGG-AATATCACTCCCCGGA TTTC-3' sg_cdkn1a_2: 5'-AAAC-GAAATCCGGGGAGTGATATT-3'. Synthesis of this sgRNA was done as described above.

## Microinjections into *N. furzeri* oocytes

For microinjections, an injection mold (manufactured by GT-Labortechnik, Arnstein, Germany) was used to form single slots of the size of an *N. furzeri* embryo. Injection plates were freshly prepared by dissolving 1.5 g of agarose in 50 ml of 0.3 x Danieau's medium. This solution was boiled in a microwave oven and then poured into a petri dish (94 mm × 16 mm). The injection mold was inserted while the solution was still liquid. As soon as the agarose has hardened (after approximately 30 min), the stamp can be removed and the plate is ready to use. Fertilized embryos were individu-ally inserted into the slots with the first cell facing the injection needle. 0.3 x Danieau's medium was added to the plate until the eggs were completely covered. The injection was performed under a stereomicroscope using glass capillary needles, a pressure injector (World Precision Instruments), and a micromanipulator (Saur). The injection solution for the inactivation of target genes contained the sgRNAs (30 ng/μl each), *Cas9* mRNA (300 ng/μl), *GFP* mRNA (200 ng/μl), and phenol red. For knock-in approaches, the concentration of sgRNA and *Cas9* mRNA was kept the same, whereas *GFP* mRNA was reduced to 100 ng/μl and 20 ng/μl of the HDR template were added. After injec-tions, the embryos remained on the plate and were stored at 29 °C until the next day. Using a fluorescence microscope, the injected embryos were sorted into GFP-positive and GFP-negative embryos. GFP-positive embryos were transferred into single wells of a 96-well plate containing 0.3 x Danieau's medium.

## DNA sampling

DNA samples were obtained either from caudal fin biopsies or whole embryos. For fin biopsies, a small part of the caudal fin was cut off from the anesthetized fish. To extract DNA from whole embryos, the embryos were mechanically disrupted using a pipet tip. For fin, biopsy samples 100 µl and for embryos, 50 µl of NaOH (50 mM) was added to the sample and subsequently incubated at 95 °C for 45 min. Afterward, 1/10 volume of Tris-HCl (1 M, pH 8.0) was added.

## Restriction enzyme digest

Restriction enzymes were used to check for the presence of mutations in the target genes. The region flanking the potential mutation site was amplified via PCR using the following oligonucleotides: *mitfa*_fw: 5′-TGCTTCACATACGTTTGCAG-3′ *mitfa*_rv: 5′-CAAAGGTCTGAGGGCTTTCC-3′, *ltk*_fw: 5′-TGTTCTGTCACCACCCTTGT-3′ *ltk*_rv: 5′-ACACTGCTATTACCAGGTTTGAC-3′, *csf1ra*_fw: 5′-CATAGATACCGTGCAAGCCTG-3′ *csf1ra*_rv: 5′-AGCCCAGGTATGAAATCCGT-3′, *slc45a2*_fw: 5′-GGATTTGGTGTTTTGGCCCT-3′ *slc45a2*_rv: 5′-GTAACTCGGCTCTAATCGTGC-3′. For the restriction enzyme digest, 20 µl of the PCR reaction were incubated over night at 37 °C after adding 6.75 µl of ddH$_2$O, 0.25 µl of the corresponding enzyme, and 3 µl of the respective enzyme buffer (*mitfa: EcoRI*, *ltk: EcoNI*, *csf1ra: NcoI-HF, slc45a2: HypCH4III*). Samples from the control digest were analyzed on a 1% agarose gel. In case of a mutation, the restriction enzyme is prevented from cleaving the PCR fragment. As a positive control, a PCR amplicon from a wild-type fish was always included in order to verify that the restriction enzyme digest worked properly.

## High-resolution melting analysis (HRMA)

After identifying the exact mutations in the targeted gene loci (from F$_2$ generation on), genotyping was performed via HRMA as described (*Krug et al., 2023*). The following oligonucleotides were used for the HRM analysis: HRMA_*mitfa*_fw: 5′-CCTCACGAGTCTCTCTATCA-3′ HRMA_*mitfa*_rv: 5′-GCCCCATGAACCCAATATAA-3′, HRMA_*ltk*_fw: 5′-CCACAGACTCTTCCAGAAAT-3′ HRMA_*ltk*_rv: 5′-CTGATTATGAGGTGCGACTA-3′, HRMA_*csf1ra*_fw: 5′-AGTGTGTGGCTTTCAATTTG-3′ HRMA_*csf1ra*_rv: 5′-TTTCTGGTGAGTGTTTGTTA-3′. To assess the genotypes, melt curves were analyzed using the *Precision Melt Analysis* software (Bio-Rad).

## Isolation of nucleic acids

RNA isolation from fish tissues was done according to the manufacturer's protocol of the RNeasy Mini Kit (Qiagen). Tissue homogenization using ceramic beats was performed with the TissueLyser II (2 min at 30 Hz). The optional on-column DNase digestion step was included. 20 µl of DEPC H$_2$O were used for final elution. RNA isolation from FACS sorted cells was done according to the protocol of the *MagMaxTM-96 Total RNA Isolation Kit*. Isolation of RNA and DNA from whole embryos was done via phenol-chloroform extraction. Using a pipette tip, the chorion of the embryos was mechanically ruptured before 500 µl of TRIzol was added. Homogenization of the embryos was performed with ceramic beats using the TissueLyser II (2 min at 30 Hz). After incubation at RT for 5 min, 200 µl chloroform was added. Samples were then mixed for 15 s, incubated at RT for 3 min, and then centrifuged at 12,000 × g for 20 min (4 °C). The upper, aqueous phase, containing the RNA, was transferred into a fresh tube, and 1.1 volumes of isopropanol, 0.16 volumes of NaAc (2 M, pH 4.0), and 1 µl of GlycoBlue (Thermo Fischer) were added. Samples were incubated at RT for 10 min and centrifuged at 12,000 × g for 20 min (4 °C). The supernatant was removed, and the pellet was washed with 1 ml of 80% EtOH and centrifuged at 7500 × g for 10 min (4 °C). The supernatant was discarded, and the pellet was air-dried. The RNA pellet was dissolved in 20 µl of DEPC-H$_2$O and stored at –80 °C. The inter- and organic phases were used for the extraction of DNA (required for genotyping PCR). 300 µl of EtOH (100%) were added, samples were incubated at RT for 2–3 min and then centrifuged at 2000 × g for 5 min (4 °C). The supernatant was discarded, and the pellet was incubated in 1 ml of 0.1 M sodium citrate in 10% EtOH for 30 min before centrifugation at 2000 × g for 5 min (4 °C). The supernatant was discarded and 1 ml of EtOH (75%) was added 15 min before centrifugation at 2000 × g for 5 min (4 °C). Pellet was air-dried and dissolved in 15 µl NaOH (8 mM).

## cDNA synthesis and gene expression analysis

For cDNA synthesis, 500 ng of RNA from fish tissues and whole embryos, 25 ng of RNA from FACS-sorted cells or 20 ng of RNA isolated from primary cells were used. cDNA synthesis was performed according to the instructions of the *iScript cDNA Synthesis Kit* (Bio-Rad). qRT-PCRs were performed in 384-well plates using 2 x SYBR Green Mix and the CFX384 Real-Time System (Bio-Rad). The reaction mix included 3 µl of cDNA (diluted 1:5 in DEPC H$_2$O), 0.4 µl of each oligonucleotide, 1.2 µl DEPC H$_2$O and 5 µl of 2 x SYBR Green Mix. Gene expression levels were determined using the following oligonucleotides: q_*mitfa*_fw: 5'-TGAAGCAAGTACTGGACAAG-3' q_*mitfa*_rv: 5'-TCCAGTAGAGTC AGAAGTCC-3', q_*ltk*_fw: 5'-CTGGGAGGAATCCGCTTA-3' q_*ltk*_rv: 5'-AGTGAGACCAGTGCAG AG-3', q_*csf1ra*_fw: 5'-AGTTCAAATGTATCAGAGACCT-3' q_*csf1ra*_rv: 5'-TATCCTGCTCCGAGAA TCAT-3', q_*gfp*_fw: 5'-AAGGGCATCGACTTCAAGGA-3' q_*gfp*_rv: 5'-GGCGGATCTTGAAGTTCACC -3', q_*ntr*_fw: 5'-CTTTTGATGCCAGCAAGAAA-3' q_*ntr*_rv: 5'-GAAGCCACAATAAAATGCCA –3', q_*cdkn1a*_fw: 5'-ATGTGCAGAGGGATGGCTAC-3' q_*cdkn1a*_rv: 5'-CCTCCAGATCTTTACGCAG-3'. For normalization the housekeeping gene *rpl13a* (q_*rpl13a*_fw: 5'-ACTGTCAGAGGCATGCTTCC-3' q_*rpl13a*_rv: 5'-TGCTCTGAAAATTGTGCGCC-3') was used.

## Whole kidney marrow (WKM) analysis via flow cytometry

Kidneys were dissected and immediately pushed through a 40 µm cell strainer (FALCON; placed on top of a 50 ml falcon tube) using a syringe plunger. Strainer and plunger were rinsed with 1 ml of PBS each. Cells were pelleted by centrifugation at 330 × g for 5 min (4 °C). The supernatant was discarded and the cell pellet was dissolved in 300 µl PBS. Analysis of the WKM was done using a BD FACS AriaTM IIIu. The gating strategy was chosen as described for the analysis of the WKM of zebrafish (*Sanz-Morejón et al., 2019*).

## Irradiation of *N. furzeri* eggs

For γ–irradiation of F$_1$ eggs from the *cdkn1a*-reporter line, eggs were placed into 12-well plates (one egg per well) containing 1 ml of 0.3 x Danieau's medium. Eggs were irradiated in the 12-well plate with a dose of 10 Gy using a Gammacell 40 Exactor (Best Theratronics Ltd.). The presence of a GFP signal was checked using an Axio Zoom.V16 with ApoTome (Zeiss).

## Genotyping of eggs and fish from the *cdkn1a*-reporter line

DNA was extracted from eggs or fin biopsies as described above. To check for the presence of the reporter construct in the *cdkn1a* locus, the following primer pair was used: *cdkn1a*_insertion_fw: 5'-TATTTCTCTGGTGTTTGCCT-3' *GFP*_insertion_rv: 5'-TGATATAGACGTTGTGGCTG-3'. To discriminate between the three different genotypes, the following primer pair was used: *cdkn1a*_genotyping_fw: 5'-CTACAGATCCAGCGTCATC-3' *cdkn1a*_genotyping_rv: 5'-CCAAGAGAACCAGACAAAGA-3'. For *cdkn1a*$^{+/+}$ animals, an amplicon with a size of 393 bp was expected, whereas a 1893 bp fragment occurred in *cdkn1a*$^{ki/ki}$ fish. In *cdkn1a*$^{ki/+}$ animals, both amplicons were expected. Genotyping PCRs were performed with the *DreamTaq* polymerase and 10 x DreamTaq Green buffer (Thermo Scientific). Cycling conditions were set according to the manufacturer's manual with an annealing temperature of 60 °C and a cycle number of 35.

## Analysis of GFP-positive cells from the *cdkn1a*-reporter line

Kidneys were removed from freshly euthanized fish and placed into 1 x PBS on ice. Together with 1 ml PBS (with 1% FBS), the tissue was pushed through a 40 µm cell strainer (FALCON), using a syringe plunger. The cell strainer and plunger were rinsed twice with 1 ml of PBS each. After centrifugation at 250 × g for 5 min (4 °C), the pellet was resuspended in 200 µl PBS containing 1% FBS. The cell suspension was stained with 5 nM Sytox Red Dead Cell Stain (Invitrogen) to identify dead cells and was subsequently analyzed on a BD FACS AriaIIIu Cell Sorter using a 100 µm Nozzle. After doublet and dead cell exclusion, the GFP-positive cells were sorted in PBS. The GFP-positive gate was set according to the negative control obtained from the kidney of a *klara* fish. Data were analyzed using FlowJo v10 Software (BD).

## Immunofluorescence staining and imaging

Liver tissues were fixed overnight in 4% paraformaldehyde in PBS and subsequently washed three times in PBS-T (0.2% Tween in PBS). Tissues were placed for 5 min in 5% sucrose, then for 2 hr in

20% sucrose, and subsequently overnight in 30% sucrose. Tissues were embedded and shock frozen in molds filled with NEG-50 cryosection medium (Thermo Scientific). The tissue was cut into 20 μm slices. After defrosting for 25 min, the sections were washed four times for 10 min at RT in PBS-T (0.2% Tween) and were permeabilized by a short washing step with permeabilization solution (0.1% Tween, 0.3% Triton X 100 in PBS). Immediately afterward, the sections were washed again two times for 10 min at RT in PBS-T (0.2% Tween) and incubated in blocking buffer (2% BSA and 10% NGS in PBS-T (0.2% Tween)) for 1 hr at RT. Afterward, the samples were incubated overnight at 4 °C with an anti-GFP antibody (Thermo Scientific Inc, United States: A-11122, rabbit) diluted 1:200 in a blocking buffer. Sections were then washed four times for 10 min at RT in PBS-T (0.2% Tween) prior to incubation with bisbenzimide Hoechst 33258 and a secondary anti-rabbit Alexa Fluor 546 antibody (Thermo Scientific Inc, United States: A-11071, goat) diluted 1:500 in blocking solution for 1 hr at RT. After multiple washing steps in PBS-T (0.2% Tween), the slides were mounted with 70 μl ProLong Diamond antifade reagent (Thermo Scientific Inc). Before imaging, the samples were incubated at 4 °C overnight. Image stacks were recorded as optical sections with the Axio Imager 2 equipped with an ApoTome.2 slider (Zeiss, Germany). The ZEN 3.4 software (Zeiss, Germany) was used to process the images and to create extended depth of focus projections of the acquired z-stacks.

## Isolation and cultivation of skin cells from *cdkn1a*$^{ki/ki}$ *klara* fish

Fish were euthanized with an overdose of Tricaine (1 g/l) and then immediately wiped with a paper towel containing 70% EtOH. After decapitation, the fish was briefly immersed three times in 0.16% NaOCl/PBS. Subsequently, scales and fins were removed with a scalpel. The skin was then pulled off using forceps, cut into small pieces, and transferred into 6 ml of HBSS (Gibco) containing 2 mg/ml collagenase type I (Fisher Scientific). After incubation for 2–3 hr with constant shaking of the reaction tubes every 15 min, the samples were centrifuged at 300 × g for 5 min followed by two wash steps using HBSS. The cells were seeded into cell culture flasks (50 ml; Greiner Bio-One). DMEM glucose (Gibco) supplemented with 10% FBS (Sigma), 0.1% MEM (1 mM; Sigma), 0.2% β-mercapto-ethanol (50 mM; Fisher Scientific), and 1 μg/ml Rh FGF (Immunotools) was used as basic medium. For the first eight days, this medium was additionally supplemented with 0.75% Pen./Strep. (Sigma), 1% Gentamicin (10 mg/ml; Sigma), and 0.3% Amphotericin B (0.25 mg/ml; Sigma). Afterward, the basic medium was supplemented with 1% Pen./Strep., 0.15% Gentamicin (10 mg/ml), 0.2% Amphotericin B (0.25 mg/ml). After 1–2 weeks, the composition of antibiotic and antifungal supplements changed to 1% Pen./Strep., 0.15% Gentamicin (10 mg/ml), and 0.1% Amphotericin B (0.25 mg/ml). As soon as 80% confluency was reached, skin cells were cultivated in a basic medium containing 1% Zellshield (Minerva Biolabs). The cells were constantly kept at 28 °C and 5% $CO_2$.

## Treatment of cells with metronidazole (Mtz)

24-well plates containing fibroblasts from the *cdkn1a*-reporter line (only *cdkn1a*$^{ki/ki}$) were used. The medium was replaced by fresh medium containing either 0.2% DMSO (control) or 50 μM of Mtz (diluted in DMSO). Immediately afterward, cells were exposed to a γ–irradiation dosage of 10 Gy using a Gammacell 40 Exactor (Best Theratronics Ltd.). Non-irradiated cells served as controls. After the irradiation, cells were kept at 28 °C and 5% $CO_2$. After 24 hr, the medium was removed and 350 μl lysis buffer RLT, supplemented with β-mercapto-ethanol (10 μl/ml buffer) from the RNeasy Mini Kit (Qiagen) was added. The cell lysate was collected and homogenized using QIAshredder spin columns. Isolation of RNA was performed according to the manufacturer's protocol. The optional on-column DNase digestion step was included. 20 μl of DEPC H2O were used for final elution.

## Sample preparation for light sheet microscopy

Throughout the preparation and the whole *in vivo* imaging process fish were constantly kept under anesthesia (200 mg/l Tricaine in system water). Depending on the age (and thus size), fish were mounted in the sample chamber either by embedding in a capillary with agarose (4 dph) or by fixing a small part of the caudal fin onto a capillary (17 dph).

The trunk and fins of fish at the age of 4 dph were completely embedded in 1% low melting agarose (in system water containing 200 mg/l Tricaine) tempered to 37 °C. The head and gills were not covered in agarose. Subsequently, the fish were sucked into a capillary tube with an inner diameter of 2.19 mm by pulling a plunger (with teflon tip) fitting into the capillary. Capillaries and matching plungers were

provided in the Chamber & Sample Holder Starter Kit Lightsheet Z.1 (Zeiss). The capillary with the fish was inserted into the sample holder and placed into the sample chamber filled with anesthesia solution (200 mg/l Tricaine in system water). After the agarose has solidified, the part of the agarose containing the fish was pushed out of the capillary (by the plunger) and the animal was imaged.

At the age of 17 dph, fish were too big to fit into the capillary. For this reason, a surface (diameter of about 3 mm) was modeled at the end of a capillary using a UV light curing resin (Bondic UV repair system). This modified capillary was placed into the sample holder and inserted into the microscope. Using the Bondic UV repair system a small part of the caudal fin was glued onto the shaped surface at the capillary. To prevent the fish from drying out, the sample chamber was then immediately filled with the anesthesia solution.

In order to ensure the vitality of the animals throughout the imaging procedure, the blood flow was constantly observed. Immediately after the images were acquired, the still anesthetized fish were euthanized with an overdose of Tricaine (1 g/l in system water).

### *In vivo* light sheet imaging and image processing

Intravital imaging was performed with a Light sheet microscope (Z1, Zeiss) enabled for dual-side illumination. It was equipped with a 20 x detection objective (W Plan-Apochromat, numerical aperture of 1.0) and an sCMOS pco. edge 4.2 camera. Furthermore, the setup was temperature-controlled by the TempModule S1 and the TempModuleCZ-LSFM in combination with the PeltierBlock S and a Temperature Sensor, which were both assembled with the sample chamber. This allowed imaging at the optimal temperature for the fish (26 °C). Image processing, including dual side fusion, brightness/contrast adjustment, and image export was conducted using the Zen 3.1 software (blue edition, Zeiss). The 3Dxl rendering module (powered by arivis) was employed for three-dimensional reconstructions of the recorded z-stacks. Conversion of z-stacks into movies was performed using the arivis Vision4D software (arivis AG).

### Statistical analysis

Data were analyzed depending on the experimental setup via t-test or One-Way ANOVA followed by Tukey's post hoc test. Equal or unequal variance was determined via F-Test followed by a Student's or Welch's t-test, respectively. The Kaplan-Meier curves (lifespan analysis) were analyzed via the Log-rank test. Significant changes are indicated by * if $p \leq 0.05$, ** if $p \leq 0.01$, and *** if $p \leq 0.001$.

## Acknowledgements

We thank Hanna Reuter for suggesting the name *klara*, Nils Hartmann for the movie on wild-type killifish mating, Annekatrin Richter for providing the picture on oocyte injection, and Hakar Aliyas, Caglar Avci, Michelle Burkhardt, Christina Ebert, Gabriele Günther, Maleen Hofmann, Erik Hüttenrauch and Dagmar Kruspe for technical support. We are very grateful to members of FLI's killifish facility, most notably to Simone Dunkel, Martin Neumann, Marcus Schmidt, Uta Naumann, and Beate Hoppe. We also thank members of the Core Facility Flow Cytometry, namely Johanna Schleep, Simone Tänzer, and Katrin Schubert for their contribution. This project was made possible by funding from the Carl Zeiss Foundation in the context of the IMPULS consortium (project number P2019-01-006) to CE. and a fellowship from the Leibniz Graduate School on Ageing and Age-Related Diseases (LGSA) to JK. The FLI is a member of the Leibniz Association and is financially supported by the Federal Government of Germany and the State of Thuringia.

## Additional information

### Funding

| Funder | Grant reference number | Author |
| --- | --- | --- |
| Carl Zeiss Stiftung | IMPULS P2019-01-006 | Christoph Englert |
| Leibniz Institute on Aging | Graduate Student Fellowship | Johannes Krug |

| Funder | Grant reference number | Author |
|---|---|---|

The funders had no role in study design, data collection and interpretation, or the decision to submit the work for publication.

## Author contributions

Johannes Krug, Conceptualization, Data curation, Formal analysis, Investigation, Writing – original draft, Writing – review and editing; Birgit Perner, Hanna Mörl, Investigation, Writing – review and editing; Carolin Albertz, Data curation, Investigation, Writing – review and editing; Vera L Hopfenmüller, Data curation, Investigation, Methodology, Writing – review and editing; Christoph Englert, Conceptualization, Supervision, Funding acquisition, Writing – original draft, Project administration, Writing – review and editing

## Author ORCIDs

Johannes Krug  http://orcid.org/0009-0000-2040-6460
Vera L Hopfenmüller  http://orcid.org/0000-0002-9807-6403
Christoph Englert  http://orcid.org/0000-0002-5931-3189

## Ethics

All fish were maintained in the Nothobranchius facility of the Leibniz Institute on Aging - Fritz Lipmann Institute Jena according to the German Animal Welfare Law. The performed experiments reported here were covered by the animal license FLI-17-016, FLI-20-001, and FLI-20-102, which were approved by the local authorities (Thüringer Landesamt für Verbraucherschutz).

## Decision letter and Author response

Decision letter https://doi.org/10.7554/eLife.81549.sa1
Author response https://doi.org/10.7554/eLife.81549.sa2

# Additional files

## Supplementary files

• MDAR checklist

## Data availability

All data generated or analysed during this study are included in the manuscript and supporting files; Source Data files have been provided for all displayed items.

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
