## [Editor Report]

This important work significantly simplifies our ability to observe and manipulate aging and senescence in a living vertebrate model. The evidence supporting the conclusions is compelling, with advanced genome editing, physiological assays, and state-of-the-art microscopy. The work will be of broad interest to biomedical and aging researchers, as well as to cell biologists.

---

## [Decision Letter]

**Decision letter after peer review:**

Thank you for submitting your article "Generation of a transparent killifish line through multiplex CRISPR/Cas9-mediated gene inactivation" for consideration by *eLife*. Your article has been reviewed by 3 peer reviewers, and the evaluation has been overseen by a Reviewing Editor, Itamar Harel, and Didier Stainier as the Senior Editor. The following individuals involved in the review of your submission have agreed to reveal their identity: Joachim Wittbrodt (Reviewer #1); Wang Wei (Reviewer #2); Martin Reichard (Reviewer #3).

Essential revisions:

1. HDR-mediated integration: as there are concerns about PCR biases in the detection of HDR-mediated integration, the authors should use another pair of primers, and describe in detail the PCR amplification. Alternatively, it would be best to use Southern Blot.

2. Quality of figures: figure quality and assembly should be improved. This includes a symmetrical representation of both males and females at all steps, a single figure that describes the generation of the Klara line, the layout of figure S3, and clearer Figure 3F.

3. Kalra line: quantification of general physiology concerning health, growth rate, and survival of this triple mutation are required.

4. p21-GFP reporter line: to demonstrate that this line could be used as a resource for investigating adult biology, in-vivo imaging of either irradiated or young/old fish (2 time points) should be performed. This should then be followed by clearance of senescent cells.

*Reviewer #1 (Recommendations for the authors):*

In the analysis of the integration, the authors should provide the exact PCR conditions, in particular the number of cycles. If they consider the statement of the perfect integration essential, this would be best supported by a Southern Blot analysis.

*Reviewer #2 (Recommendations for the authors):*

Concerns:

1) Figures in this manuscript are not well-organized. For instance, it is difficult to find the phenotype for male and female Klara fish. In some figures, the authors show images of males, while in other figures I can only find females. Three figures were used to demonstrate the establishment of the Klara fish line, which is not necessary. One or two are more than enough. More figures should be used to characterize the Klara fish line. For example, a detailed characterization of this line during aging should be provided because the authors claim that this line will be useful for aging studies. Does the triple mutation influence the speed of aging? Would fish develop slower, faster, or normally? It is important to have these data because it will determine, to what extent, we can use the line for aging-, development-, and regeneration-related studies.

2) The crystal-clear mutants have been established in zebrafish; can authors generate a truly crystal-clear animal in African killifish? The re-appearance of pigmentation limits the application of this line in the adult stages.

3) One of the key goals of this manuscript is to establish a tool for *in vivo* imaging. But the authors did not use the important lines they have established to perform *in vivo* imaging. The cdkn1a (p21)-GFP line is very interesting, it can be informative if authors track GFP expression in the skin during aging for a single individual for a month or two. Imaging can be done weekly.

---

## [Author Response]

Essential revisions:1. HDR-mediated integration: as there are concerns about PCR biases in the detection of HDR-mediated integration, the authors should use another pair of primers, and describe in detail the PCR amplification. Alternatively, it would be best to use Southern Blot.

We fully understand and acknowledge that there are concerns regarding the specific integration of the GFP-NTR cassette into the killifish genome, which in the initial submission has only been demonstrated by PCR using a single primer pair. We have now used a second pair of primers to verify the site-specific integration of the cassette. The corresponding data are shown in Figure 5—figure supplement 1B. We would also like to point out that the ‘functionality’ of the cassette strongly suggests its proper integration.

2. Quality of figures: figure quality and assembly should be improved. This includes a symmetrical representation of both males and females at all steps, a single figure that describes the generation of the Klara line, the layout of figure S3, and clearer Figure 3F.

We have improved both the quality and assembly of the figures. This includes a balanced representation of males and females, the combination of all data regarding generation of the klara line in one figure and improving the quality/layout of figure 3. We have expanded the latter, which now also contains the scheme of former figure S3. We assume “3F” actually means “2F”. Here, we have added a female animal.

3. Kalra line: quantification of general physiology concerning health, growth rate, and survival of this triple mutation are required.

We have now included information on health, growth rate, and survival of the klara line. First, we have included a ‘survival’ curve of klara and corresponding wild-type animals (Figure 2—figure supplement 1A). Of note, this is not a dedicated survival curve but a compilation of data on the daily records from the fish facility regarding animals found dead. There is no significant difference between the two cohorts. Second, we have quantified size and weight until 5 weeks post-hatching in both cohorts (Figure 2—figure supplement 1B). Third, all fish in our fish facility are inspected and scored daily regarding swimming, feeding and social behavior as well as body condition, appearance and breathing. There is no parameter in which a difference between klara and wild-type animals have been observed. This is mentioned in the revised manuscript.

4. p21-GFP reporter line: to demonstrate that this line could be used as a resource for investigating adult biology, in-vivo imaging of either irradiated or young/old fish (2 time points) should be performed. This should then be followed by clearance of senescent cells.

We fully agree with the reviewers that the demonstration of the possibility of *in vivo* microscopy of klara animals carrying the GFP allele in the cdkn1a locus significantly adds to the impact of our work. We have performed light sheet microscopy on living klara animals at two different time points, namely at 4- and 17-days post-hatching and have included respective images and movies into the manuscript (Figure 5J-M and Figure 5-video1 and video2).

We would like to point out that both reviewer requests pertain animal experiments for which, at least initially, we did not have the appropriate licenses. Regarding microscopy we have applied for an amendment of an existing license, which made it possible to perform the light sheet experiments, albeit at restricted time points. Regarding the clearing experiment, however, we did not have any pre-existing license. Therefore, we performed those experiments in cell culture. The respective data are shown in Figure 5—figure supplement 2C.

Reviewer #1 (Recommendations for the authors):In the analysis of the integration, the authors should provide the exact PCR conditions, in particular the number of cycles. If they consider the statement of the perfect integration essential, this would be best supported by a Southern Blot analysis.

To show the integration of the reporter cassette into the cdkn1a locus, we initially performed a PCR using a forward primer that binds in the cdkn1a locus, but outside the sequence covered by the flanking arm, and a reverse primer binding within the GFP sequence. With this PCR, an amplicon was only obtained upon integration of the construct in the cdkn1a locus. In addition, we included an alternative PCR using a different primer pair combination (shown in in Figure 5—figure supplement 1B). With both primer pairs it was possible to verify the presence of the reporter cassette. In addition, the second primer pair also allows to distinguish between cdkn1a^+/+^, cdkn1a^ki/+^ and cdkn1a^ki/ki^ fish and is thus suitable for genotyping of offspring.

The PCRs were performed using DreamTaq as Polymerase. The general PCR conditions were selected according to the manufacturer’s manual. We used 60°C as an annealing temperature and 35 cycles per PCR. Those conditions were included in the Methods and Materials section.

Reviewer #2 (Recommendations for the authors):Concerns:1) Figures in this manuscript are not well-organized. For instance, it is difficult to find the phenotype for male and female Klara fish. In some figures, the authors show images of males, while in other figures I can only find females. Three figures were used to demonstrate the establishment of the Klara fish line, which is not necessary. One or two are more than enough. More figures should be used to characterize the Klara fish line. For example, a detailed characterization of this line during aging should be provided because the authors claim that this line will be useful for aging studies. Does the triple mutation influence the speed of aging? Would fish develop slower, faster, or normally? It is important to have these data because it will determine, to what extent, we can use the line for aging-, development-, and regeneration-related studies.

We agree with the reviewer that a single figure about the generation of the klara line is more suitable. For this reason, Figure 1 and supplementary figure 1 summarize the line generation. We extended the characterization of klara fish by including a lifespan analysis (compared to the genetic background strain MZCS-08/122). It is important to mention, that this was not a dedicated lifespan experiment but a compilation of survival data obtained from our fish database (date of hatching vs date of death). Only fish that were found dead were included in the analysis, which revealed no difference in the survival of klara fish compared to respective wild-types (Figure 2—figure supplement 1A). In addition, we also included data concerning body size and weight of male and female klara as well as wild-type fish (Figure 2—figure supplement 1B) showing that klara fish develop normally.

2) The crystal-clear mutants have been established in zebrafish; can authors generate a truly crystal-clear animal in African killifish? The re-appearance of pigmentation limits the application of this line in the adult stages.

We believe that it is in general possible to establish a fully transparent N. furzeri line by selecting different target genes. A potential inactivation of other genes involved in the formation of melanophores, such as tyrosinase or slc45a2, could be beneficial. In the meantime, the mutation causing the lack of iridophores in the casper line has been identified (D’Agati et al., 2017), so that an inactivation of mpv17 in N. furzeri could also be tested. However, these experiments are outside of the scope of this manuscript.

3) One of the key goals of this manuscript is to establish a tool for in vivo imaging. But the authors did not use the important lines they have established to perform *in vivo* imaging. The cdkn1a (p21)-GFP line is very interesting, it can be informative if authors track GFP expression in the skin during aging for a single individual for a month or two. Imaging can be done weekly.

The aim to use the klara line for in vivo imaging, in form of the cdkn1a-reproter line, has been addressed now and data were included in Figure 5J-M and corresponding movies. We were able to get the approval for an animal license allowing us the *in vivo* imaging of living, anesthetized fish using light sheet microscopy at two different timepoints (4- and 17dph). With this technique, we were able to detect GFP-positive cells in cdkn1a^ki/ki^ fish at both timepoints. Of note, the number of GFP-positive cells significantly increased from 4dph to 17dph.